# Characteristics of Lactic Acid Bacteria as Potential Probiotic Starters and Their Effects on the Quality of Fermented Sausages

**DOI:** 10.3390/foods13020198

**Published:** 2024-01-08

**Authors:** Yinchu Liu, Sai Gao, Yue Cui, Lin Wang, Junya Duan, Xinyu Yang, Xiaochang Liu, Songshan Zhang, Baozhong Sun, Haojie Yu, Xiaoguang Gao

**Affiliations:** 1College of Food Science and Biology, Hebei University of Science and Technology, Shijiazhuang 050018, China; liuyinchu88@163.com (Y.L.); gaosai080911@163.com (S.G.); w592148022@163.com (L.W.); jy19970510@163.com (J.D.); yxinyu1127@163.com (X.Y.); 2Institute of Animal Sciences, Chinese Academy of Agricultural Sciences, Beijing 100193, China; lxc_cau@163.com (X.L.); zhangsongshan_1997@163.com (S.Z.); baozhongsun@163.com (B.S.); haojieyu815@163.com (H.Y.)

**Keywords:** fermented sausage, lactic acid bacteria, probiotic, antimicrobial, antioxidant

## Abstract

The aim of this study was to explore the potential of commercial lactic acid bacteria (LAB) as probiotic starters in fermented sausages. We initially investigated the growth activity, acid production capability, and tolerance to fermentation conditions of *Lactobacillus sakei*, *Lactiplantibacillus plantarum*, and *Pediococcus pentosaceus*. All three LAB strains proved viable as starters for fermented sausages. Subsequently, we explored their potential as probiotics based on their antibacterial and antioxidant capabilities. *L. plantarum* exhibited stronger inhibition against *Escherichia coli* and *Staphylococcus aureus*. All three strains displayed antioxidant abilities, with cell-free supernatants showing a higher antioxidant activity compared to intact cells and cell-free extracts. Moreover, the activities of superoxide dismutase, glutathione peroxidase, and catalase were stronger in the cell-free supernatant, cell-free extract, and intact cell, respectively. Finally, we individually and collectively inoculated these three LAB strains into sausages to investigate their impact on quality during the fermentation process. External starters significantly reduced pH, thiobarbituric acid reactive substances, and sodium nitrite levels. The improvements in color and texture had positive effects, with the *L. plantarum* inoculation achieving higher sensory scores. Overall, all three LAB strains show promise as probiotic fermentation starters in sausage production.

## 1. Introduction

In recent years, there has been a growing interest in the relationship between food and health, with demand for food extending beyond mere taste and nutritional value. Food has evolved to serve as a proactive tool for improving human health through additional functional properties, leading to a gradual increase in the demand for functional foods [1]. Presently, a significant advancement in functional foods is associated with those containing probiotic cultures [2]. Probiotics possess essential functional attributes that can fulfill a vast majority of our fundamental nutritional and therapeutic supplement requirements. It is anticipated that the probiotics market will reach a valuation of 95.25 billion dollars by 2028 [2]. For probiotics to exert beneficial effects on health and well-being, they must be alive and consumed in adequate quantities [3]. Therefore, they must survive acidic conditions in the stomach and resist bile acids at the outset of the small intestine [4]. Sausages present an advantageous avenue for transporting probiotic cells into the intestine as they are encapsulated within the sausage matrix comprising meat and fat. This encapsulation greatly enhances the survival of probiotic cultures during the critical passage through the stomach and small intestine compared to their exposure without protection to low pH and bile salts [5].

Fermented sausages encompass a range of traditional meat products that enjoy widespread popularity in numerous countries worldwide. The diversity arising from varied raw materials, ingredients, and manufacturing processes results in distinct flavors, textures, and shapes. Compared to natural fermentation, the use of artificially inoculated starters ensures the consistent safety and quality of the products [6,7]. The primary fermentation starters include the genera of *Pediococcus*, *Lactobacillus*, *Staphylococcus*, and *Debaryomyces* [8,9]. Fermented sausages rely on the production of lactic acid and the subsequent pH reduction, making lactic acid bacteria (LAB) pivotal in their fermentation [10]. Lowering the pH effectively inhibits the proliferation of pathogens [11]. Additionally, LAB, such as *Lactobacillus* and *Pediococcus*, exhibit probiotic properties and offer beneficial effects within the gastrointestinal tract [12].

In the routine metabolic activities of aerobic organisms, reactive oxygen species (ROS) naturally emerge, serving pivotal functions in immune responses against invading microorganisms and regulating communication between cells [13]. However, excessive ROS production can lead to oxidative stress, resulting in protein oxidation, lipid peroxidation, and DNA damage [14]. An excess of ROS in the intestine causes oxidative stress, contributing to various intestinal diseases, inflammation, and functional impairments [15]. It has been reported that certain LAB possess antioxidant capabilities [16]. Studies have demonstrated that LAB isolates possess the ability to scavenge free radicals such as 1,1-diphenyl-2-picrylhydrazyl (DPPH) and 2,2′-azino-bis (3-ethylbenzothiazoline-6-sulfonicacid) (ABTS) [17,18]. Certain studies suggest that the antioxidant capabilities of specific LAB strains might be associated with antioxidative enzymes, such as superoxide dismutase (SOD), catalase (CAT), and glutathione peroxidase (GSH-PX) [19,20]. Interest in the antioxidative capacity of LAB is on the rise. Although extensive global research has been conducted on the antioxidant potential of LAB both in vitro and in vivo, the mechanisms regulating resistance to oxidative stress remain inadequately explored.

In this study, we selected three commercially available LAB strains—*Lactobacillus sakei*, *Lactiplantibacillus plantarum*, and *Pediococcus pentosaceus*—to investigate their antibacterial activity and antioxidant properties, assessing their potential as fermentation starters in sausage production. Subsequently, we individually and collectively inoculated these three LAB strains into fermented sausages to determine their impact on the quality of the fermented sausages.

## 2. Materials and Methods

### 2.1. Materials

Fresh lean pork and pork back fat were purchased from Shuangge Food Co., Ltd. (Shijiazhuang, China). *Lactobacillus sakei*, *Lactiplantibacillus plantarum*, and *Pediococcus pentosaceus* were obtained from the China Industrial Microbial Culture Collection Center (Beijing, China). *Escherichia coli* and *Staphylococcus aureus* were provided by Hebei University of Science and Technology (Shijiazhuang, China). All necessary raw materials for sausage production, such as casings, sodium nitrite, sodium tripolyphosphate, salt, glucose, sucrose, and white pepper powder, were purchased from Beiguo Supermarket (Shijiazhuang, China). MRS liquid and solid media, as well as LB solid and liquid media, were acquired from OboStar Biotechnology Co., Ltd. (Beijing, China). All employed chemical reagents were of analytical grade.

### 2.2. Growth Activity and Acid-Producing Capacity of LAB

After activation in MRS liquid medium, *L. sakei*, *L. plantarum*, and *P. pentosaceus* were inoculated at a 1% (*v*/*v*) inoculum size into fresh MRS liquid medium. The cultures were then incubated at 37 °C for 24 h. The OD600 and pH of the cultures were measured every 2 h.

### 2.3. Tolerance of LAB

After adjusting the pH values to 4.0, 5.0, and 6.0, and varying the concentrations of NaCl (0%, 2%, and 3%) (*w*/*v*) and NaNO_2_ (0, 75, and 150 mg/kg) in the MRS liquid media, the previously activated *L. sakei*, *L. plantarum*, and *P. pentosaceus* were inoculated at a 1% (*v*/*v*) inoculum size into the MRS liquid media and incubated at 37 °C for 24 h. The OD600 of the cultures were measured after 24 h.

### 2.4. Antimicrobial Activity of LAB

The previously activated *L. sakei*, *L. plantarum*, and *P. pentosaceus* were inoculated at a 1% (*v*/*v*) inoculum size into MRS liquid medium at 37 °C for 24 h. After incubation, the cultures were centrifuged (8000× *g*, 5 min, 4 °C), and the resulting supernatants were collected. For the Oxford cup agar diffusion method, *S. aureus* and *E. coli* were used as indicator bacteria. They were first activated in LB liquid medium and subsequently poured into unconsolidated LB solid media at different pH values (4.0, 5.0, 6.0, and 7.0) at 45 °C. After the media solidified, Oxford cups were placed, and 200 μL of the supernatant was added. As a blank control, 200 μL of sterile MRS liquid medium was used. The plates were subsequently incubated at 37 °C for 24 h. The measurement of the inhibition zone’s diameter was conducted using calipers.

### 2.5. Antioxidant Activity of LAB

#### 2.5.1. Preparation of Cell-Free Supernatant, Intact Cells, and Cell-Free Extracts

Cell-free supernatant, intact cells, and cell-free extracts were prepared and based on the research of Lee et al. [19]. *L. sakei*, *L. plantarum*, and *P. pentosaceus* were first activated in MRS liquid medium, and their concentrations were adjusted to 1 × 10^9^ CFU/mL. The cultures were then centrifuged (10,000× *g*, 10 min, 4 °C), and the resulting supernatant was passed through a 0.45 μm sterile filter membrane to prepare a cell-free supernatant. The cell precipitate was collected and washed with a sterile sodium phosphate-buffered solution (PBS, 0.02 mol/L, pH 7), and the bacterial concentration was adjusted to 1 × 10^9^ CFU/mL to prepare intact cells. To obtain cell-free extracts, the 1×10^9^ CFU/mL bacterial cell suspension was subjected to ultrasonic disruption (including ten 1 min strokes at 1 min intervals in an ice bath), and centrifugated (10,000× *g*, 10 min, 4 °C) to remove cell debris. Subsequently, the three types of LAB samples prepared were utilized for antioxidant assays, conducting three repetitions for each parameter.

#### 2.5.2. Ability to Neutralize 2,2-Diphenyl-1-picrylhydrazyl Free Radical

To establish the sample group, 1 mL of the sample was combined with 1 mL of a 0.2 mM 2,2-diphenyl-1-picrylhydrazyl (DPPH) solution in ethanol and kept in the dark at 25 °C for 30 min. The control and blank groups were prepared by replacing the samples with equal volumes of distilled water and ethanol, respectively. The mixture was centrifuged, and the absorbance of the supernatant was measured at 571 nm.
(1)$$\mathrm{DPPH}\;\mathrm{free}\;\mathrm{radical}\;\mathrm{scavenging}\;\mathrm{activity}\;(\%)=[1-(A_{\mathrm{Sample}}-A_{\mathrm{Blank}})/A_{\mathrm{Control}}]\times 100$$A_Sample_, A_Control_, and A_Blank_ represent the absorbance of the sample, control, and blank groups, respectively.

#### 2.5.3. Ability to Scavenge Hydroxyl Radical

To establish the sample group, 1 mL of the sample solution was mixed with 1 mL of 2.5 mmol/L ortho-phenanthroline, 1 mL of sodium PBS (0.02 mol/L, pH 7.4), and 1 mL of FeSO_4_ (9 mmol/L). The mixture was then combined with 1 mL of H_2_O_2_ (20 mmol/L) solution and incubated at 37 °C for 1 h. The control group used distilled water instead of the sample solution, and the blank group used distilled water instead of the sample solution and H_2_O_2_. The hydroxyl (OH) radical scavenging activity was determined by recording the absorbance of each of these three groups at 536 nm, calculated using the following equation:(2)$$\mathrm{OH}\;\mathrm{radical}\;\mathrm{scavenging}\;\mathrm{activity}\;(\%)=(A_{\mathrm{Sample}}-A_{\mathrm{Control}})/(A_{\mathrm{Blank}}-A_{\mathrm{Control}})\times 100$$A_Sample_, A_Control_, and A_Blank_ represent the absorbance of the sample, control, and blank groups, respectively.

#### 2.5.4. Activity to Scavenge 2,2′-Azino-bis (3-ethylbenzothiazoline-6-sulfonic acid) Radicals

The assay to determine the 2,2′-azino-bis (3-ethylbenzothiazoline-6-sulfonic acid) (ABTS) cation was carried out employing a modified method outlined in a prior study [21]. First, 2.5 mL of ABTS^+^ solution (7 mmol/L) was combined with 44 μL of K_2_S_2_O_8_ (140 mmol/L), allowing the mixture to react at 25 °C in the absence of light for 16 h. After the reaction, the mixture was diluted with sodium PBS (20 mmol/L, pH = 7.4) until the OD734 value reached 0.700, resulting in the preparation of an ABTS^+^ working solution. To establish the sample group, 3.0 mL of the ABTS^+^ working solution was combined with 30 μL of the sample solution, and the mixture was allowed to react at 25 °C for 6 min. Subsequently, the OD734 value was measured. The control group consisted of equal volumes of sodium PBS instead of the samples. The blank group consisted of equal volumes of sodium PBS instead of the ABTS^+^ working solution.
(3)$$\mathrm{ABTS}\;\mathrm{radical}\;\mathrm{scavenging}\;\mathrm{activity}\;(\%)=[1-(A_{\mathrm{Sample}}-A_{\mathrm{Blank}})/A_{\mathrm{Control}}]\times 100$$A_Sample_, A_Control_, and A_Blank_ are the absorbance of the sample, control, and blank groups, respectively.

#### 2.5.5. Activity to Scavenge Reducing Power

Following the method to conduct reducing power described by Wu et al. [22] with slight modifications, 0.5 mL of the sample solution was combined with 0.5 mL of sodium PBS (0.2 mol/L, pH 6.6) and 0.5 mL of potassium ferrocyanide solution (1%, *w*/*v*). The mixture was then heated at 50 °C for 20 min. After cooling to 25 °C, 0.5 mL of trichloroacetic acid solution (TCA, 10%, *w*/*v*) was added, and the mixture was centrifuged (5000× *g*, 5 min, 25 °C). One milliliter of the supernatant was collected and combined with 1 mL of distilled water and 1 mL of FeCl_3_ (0.1%, *w*/*v*). The resulting mixture underwent a reaction at 25 °C for 10 min, and the absorbance value was subsequently determined at 700 nm.

### 2.6. Antioxidant Enzyme Activity of LAB

The activities of SOD, GSH-PX, and CAT in LAB samples were assessed using commercial assay kits (Nanjing Jiancheng Bioengineering Institute, Nanjing, China), following the manufacturer’s instructions. The activities were expressed as U/mg protein.

### 2.7. Preparation of Fermented Sausages

The pork lean meat and fat (weight-to-weight ratio of 8:2) were washed, ground, and marinated for 24 h with various seasonings (Table 1). Specifically, the lean meat was derived from the pork loin, and the fat was obtained from the back fat. Following this preparation, the activated bacterial cultures were centrifuged (10,000× *g*, 10 min, 4 °C) to collect cell pellets. These pellets underwent a subsequent wash with sterile saline (0.9%, *w*/*v*) and were then resuspended to achieve a concentration of 10^9^ CFU/mL. Next, five different starters were added to the process as follows: (1) CK (no starters added), (2) LS (only *L. sakei* added), (3) LP (only *L. plantarum* added), (4) PP (only *P. pentosaceus* added), and (5) LS-LP-PP (with a 1:1:1 ratio of *L. sakei*, *L. plantarum*, and *P. pentosaceus*). The bacterial suspensions were blended with the meat at a ratio of 10 mL/kg. The bacterial count for each starter was adjusted to 10^7^ CFU/g. Subsequently, the resulting mixture was filled into collagen casings measuring 25 mm in diameter. After creating punctures in the casings using sterilized toothpicks, the sausages were placed in a fermentation incubator (RLD-450E-4, Ningbo LeDian, Ningbo, China) in a laboratory (Shijiazhuang, China). They fermented at 30 °C with 85% relative humidity for 32 h before subsequent analysis. This entire process was independently conducted in three separate batches.

### 2.8. Physicochemical Analyses

#### 2.8.1. Measurement of pH

After removing the collagen casings, 3 g of fermented sausage were homogenized (6000 rpm, 50 s) with 30 mL of KCl solution (0.1 mol/L). The pH of the mixture was determined using a pH meter (ATI Orion 420, Orion Inc., Boston, MA, USA). This process was repeated in duplicate for each of the three batches.

#### 2.8.2. Assessment of Lipid Oxidation

Lipid oxidation was evaluated by observing alterations in 2-thiobarbituric acid-reactive substances (TBARs) over the storage period, adopting the procedure outlined by Serdaroglu et al. [23]. The measurement was conducted at 532 nm, and a standard curve was generated using 1,1,3,3-tetraethoxypropane as a reference standard. The TBARS content was quantified and expressed as mg MDA/kg. Two replicates were performed for each of the three batches.

#### 2.8.3. Determination of Sodium Nitrite Content

The method of naphthylenediamine hydrochloride in the national standard GB 5009.33-2016 was used to determine the content of sodium nitrite in sausages after fermentation [24]. Three batches, each with two replicates, were analyzed.

#### 2.8.4. Color Measurement

The L* (lightness), a* (red-green component), and b* (yellow-blue component) values of 1.5 cm thick sausage slices were measured using a colorimeter (CM-5, Konica Minolta, Tokyo, Japan), following the method described by Hu et al. [25]. Two replicates were performed for each of the three batches.

#### 2.8.5. Texture Profile Analysis

Texture profile analysis (TPA) was performed using a TA-XT/Plus/50 texturometer (Stable Micro Systems, Godalming, UK), following the method described by Sirini et al. [26]. Each sample underwent two compression cycles using an aluminum cylindrical probe (p/5). The test was set to automatic triggering with pre-test, test, and post-test speeds set at 2.0 mm/s, 1.0 mm/s, and 2.0 mm/s, respectively. Hardness, defined as the force required to attain a certain deformation on the sausage surface; springiness, which refers to the speed of recovery of the deformed sausage after force removal; cohesiveness, denoting the extent of internal bonding within the sausage; and chewiness, characterized by the energy needed to masticate the sausage, were measured. Two replicates were performed for each of the three batches.

### 2.9. LAB Count

The LAB count in the fermented sausages was determined following the national standard GB 4789.35-2016 [27]. The results were reported as log_10_ CFU/g, with two replicates conducted for each of the three batches.

### 2.10. Sensory Evaluation

The color, aroma, texture, and taste of the fermented sausages were sensory-analyzed using the scoring criteria in Table 2. The sensory evaluation panel consisted of five male and five female students from the food program of Hebei University of Science and Technology. Prior to the actual evaluation, the team members received a two-week training on the sensory quality attributes of fermented sausages. Each sample was cut into 1 cm thick slices and randomly assigned to the assessors. Additionally, the mouth was rinsed with warm water while tasting the sausages between different groups, and the assessment was performed without communication among the members of the assessment team. Over three days, three batches were evaluated, resulting in a total of 30 sensory analyses for each parameter in each group.

### 2.11. Statistical Analysis

The results were expressed as the mean values ± standard deviation. Significant differences were evaluated using Duncan’s multiple comparisons within a one-way analysis of variance (ANOVA) conducted with SPSS version 26 (SPSS, IBM, Chicago, IL, USA). *p*-values below 0.05 were considered statistically significant. All figures were created using GraphPad Prism 9.4.1 (GraphPad Software INC., La Jolla, CA, USA).

## 3. Results and Discussion

### 3.1. Growth of LAB In Vitro

From Figure 1A–C, variations in the growth activities and acid production capabilities were observed among different LAB strains. After 4 h, 6 h, and 4 h, *L. sakei*, *L. plantarum*, and *P. pentosaceus* entered the logarithmic growth phase, respectively. During this phase, they exhibited the shortest division time, active enzyme systems, and vigorous metabolism. Early entry into the logarithmic growth phase is crucial for LAB to establish themselves as the dominant species, significantly contributing to the improvement of fermented sausage quality. After 20 h, with the depletion of nutrients, accumulation of harmful byproducts, or disturbances in environmental conditions, all three LAB strains entered a declining phase. As *L. sakei*, *L. plantarum*, and *P. pentosaceus* entered the logarithmic growth phase, the pH rapidly decreased, reaching 3.75, 3.61, and 3.76 after 24 h, respectively. The fast growth and acid production were responsible for inhibiting the growth of pathogenic and spoilage bacteria, ensuring the safety of the sausages during the fermentation process [28].

Bacterial growth in food matrices may be influenced by intrinsic and extrinsic factors. The selected LAB used as starters must be able to adapt to the specific conditions of the food matrix [29]. In fermented sausages, LAB strains should exhibit resistance to salt, nitrite, and acidic pH. Additionally, these strains should demonstrate quick activation and robust growth throughout the fermentation and maturation processes [30]. With the increase in NaCl content, all three LAB strains were slightly inhibited (Figure 1E). At higher NaCl concentrations, microbial cells may undergo cell wall separation, structural damage, disruption of strain physiology, slow growth, or death. Under typical conditions for fermented sausages, with a pH range of 4.6–5.3, NaNO_2_ content below 150 mg/kg, and NaCl content between 2.5–3%, all three LAB strains can grow normally (Figure 1D,E). Therefore, all three LAB strains have the potential to be used as culture starters.

### 3.2. Antibacterial Activity of LAB

The evaluation of antibacterial activity of different LAB under varying pH conditions was conducted using the Oxford cup agar diffusion method. The results indicated that the supernatant of LAB exhibited antibacterial activity within the pH range of fermented sausages (Figure 2). Regardless of the target bacteria, including *E. coli* and *S. aureus*, *L. plantarum* exhibited the strongest antibacterial activity, followed by *L. sakei* and *P. pentosaceus*. There were no significant differences in antibacterial activities under different pH conditions. The primary antibacterial mechanisms of LAB are attributed to their acid production, while their bacteriocin production also offers additional control over potential pathogens in sausages [31]. Product safety can be significantly enhanced by utilizing starters capable of reducing or inhibiting the growth of pathogenic microorganisms associated with sausages, potentially benefiting consumer health.

### 3.3. Antioxidant Activity of LAB

The formation of DPPH, ABTS, and OH radicals was inhibited by cell-free supernatants, cell-free extracts, and intact cells of three LAB (Figure 3). The cell-free supernatant from LAB has been utilized as an in vitro model for evaluating the functional effects of biologically active metabolites secreted by living bacteria [32]. In this study, cell-free supernatants exhibited stronger antioxidant activity compared to intact cells and cell-free extracts. Similar results were reported by Lee et al. [19], who isolated five strains of LAB from kimchi, with their cell-free supernatants showing higher DPPH, ABTS, and OH radical inhibition compared to intact cells and cell-free extracts. The variation in antioxidant activity is probably due to the generation of bioactive compounds throughout LAB growth. The intact cells and cell-free extracts of the three LAB strains also exhibited varying degrees of antioxidant activity. The lipid peroxidation inhibition and free radical scavenging activity are associated with the presence of polysaccharides, proteins, lipoteichoic acid, peptidoglycans, and exopolysaccharides found on the surfaces of LAB [33]. Additionally, amino acids and peptides within the cells also possess antioxidant activity. Different LAB strains displayed varying reducing power, with the cell-free supernatants exhibiting stronger reducing power compared to cell-free extracts and intact cells. Among the strains, *P. pentosaceus* exhibited the highest reducing power.

### 3.4. Antioxidant Enzyme Activity of LAB

SOD plays a vital role in cellular defense against the detrimental impacts of oxygen free radicals [34]. This metalloprotein contains metal ions in its active centers (Fe, Cu, Zn, Mn) and is involved in the conversion of the superoxide radical into oxygen and hydrogen peroxide (H_2_O_2_) molecules [35]. The SOD activity of all three LAB strains was higher in the cell-free supernatant compared to the cell-free extract and intact cell (Table 3). *L. sakei* exhibited the highest antioxidant activity, followed by *L. plantarum* and *P. pentosaceus*. However, the H_2_O_2_ generated by SOD in the dismutation reaction was a weak oxidizing agent that can easily penetrate the cell membrane, causing oxidative damage to DNA, proteins, and lipids [36]. H_2_O_2_ contributes to oxidation damage directly or as a precursor or as hydroxyl radicals. It is eliminated by GSH-PX or CAT. Among the three LAB strains, only GSH-PX activity was observed in the cell-free extract, with *L. sakei* displaying the highest activity, followed by *P. pentosaceus* and *L. plantarum*. The cell-free supernatant, intact cells, and cell-free extract of three LAB strains all exhibited CAT activity, with intact cells showing the highest activity. The antioxidant mechanisms of probiotic LAB are intricate and differ across various strains [19].

### 3.5. Changes in Physicochemical Parameters and LAB Counts of Fermented Sausage

A remarkable viability of LAB was observed throughout the entire fermentation period in both the microbial inoculation groups and the control group. At the end of the fermentation, LAB counts in both the inoculated and control groups ranged from 7.72 log_10_ CFU/g to 9.25 log_10_ CFU/g (Figure 4B). All treatment groups of sausages exhibited a decreasing trend in pH during the fermentation process, with LS, LP, PP, and LS-LP-PP showing significantly greater reductions in pH compared to CK (Figure 4A). The decline in pH is attributed to the production of lactic acid by LAB during carbohydrate metabolism [37]. This aligns with the gradual increase in LAB depicted in Figure 4B. The reduction in pH during sausage fermentation is a critical requirement as it contributes to the inhibition of undesirable foodborne microorganisms and accelerates the development of the red color [38]. A lower pH value plays a pivotal role in shaping the unique flavors, colors, aromas, and microbial consistency of fermented sausages [39].

TBARS is a suitable indicator for assessing lipid oxidation throughout the fermentation process, primarily utilized to measure the formation of secondary products such as malonaldehyde, aldehydes, ketones, and other oxidative by-products. Lipid oxidation in meat products can result in alterations to color, flavor, and nutritional content, and may pose health hazards [40]. In the different treatment groups, TBARS levels increased during the fermentation process. The microbial inoculation groups exhibited significantly lower TBARS levels than the control group, indicating the antioxidative properties of all four starters. Notably, the mixed inoculation group had lower TBARS levels compared to single inoculation (Figure 4C). This suggests that the mixed inoculation has a greater capacity to inhibit lipid oxidation, as previously reported by Chen et al. [41].

The use of sodium nitrite in fermented sausages inhibits microbial growth, delays spoilage, stabilizes the red color of the meat, and contributes to the distinctive flavor of sausages. However, excessive sodium nitrite, under heating conditions, can react with biogenic amines in the meat to form N-nitrosamines, which are potent carcinogens harmful to human health [42,43]. At the end of fermentation, the sodium nitrite content in all treatment groups fell within a range of 4.53 mg/kg to 8.52 mg/kg, all of which were below safety thresholds (Figure 4D). Throughout the fermentation process, the sodium nitrite content decreased in all treatment groups, with significantly lower levels in the microbial inoculation groups compared to the control group. This is consistent with findings by Zhang et al. [44], who reported that *L. curvatus* and *P. pentosaceus* isolated from Chinese Dong sausages also reduced nitrite content in fermented sausages.

The microbial inoculation groups exhibited a significant increase in the L* and a* values of the fermented sausages compared to the control group, accompanied by a significant decrease in the b* value (Table 4). The chemical reduction of nitrite results in the formation of nitric oxide (NO), which reacts with myoglobin to produce nitrosomyoglobin, contributing to the typical bright red coloration [45]. The effect of LAB inoculation on the L* and a* values of the fermented sausages is consistent with the changes in sodium nitrite content. According to Sirini et al., the generation of lactic acid during the fermentation process can lead to an increase in the L* and a* values [26]. Compared to naturally fermented sausages, sausages fermented with added LAB starters showed a significant improvement in color.

The results of the texture profile analysis (TPA) are presented in Table 4. The hardness, springiness, cohesiveness, and chewiness of microbial inoculation groups were significantly higher than those of the control group (*p* < 0.05). During fermentation, pH declines and myofibrillar proteins aggregate into gels, resulting in changes in the structure of the sausage [46].

### 3.6. Sensory Evaluation Analysis

The sensory attributes (color, odor, texture, taste, and total scores) of fermented sausages were assessed by the trained panel after the completion of the fermentation process, as depicted in Table 5. The color, odor, and taste of the microbial inoculation groups and the control group showed no significant differences. The highest scores of texture and total were observed in the LP group, with no significant differences between the other inoculation groups and the control group. Overall, the LS, PP, and LS-LP-PP groups showed no significant changes in sensory characteristics compared to naturally fermented sausages. The incorporation of *L. plantarum* significantly enhanced the texture of the fermented sausages, making it a potential candidate for future fermented sausage starter cultures.

## 4. Conclusions

In conclusion, the three selected commercial LAB strains—*L. sakei*, *L. plantarum*, and *P. pentosaceus*—demonstrate potential as probiotic starters for fermented sausages, showcasing both antibacterial and antioxidant activities. These LAB strains exhibited notable inhibition against *E. coli* and *S. aureus*. Furthermore, they exhibited DPPH, ABTS, and OH free radical scavenging activities, along with strong reducing power and antioxidant enzyme activities. The use of probiotic starters not only maintained the physicochemical attributes and sensory evaluation of naturally fermented sausages but also enhanced the products’ color and texture. In particular, higher sensory scores were achieved through the inoculation with *L. plantarum*. Probiotic fermented sausages not only introduce innovation into the meat industry but also offer consumers a new way to consume probiotics, serving as a promising option for health-conscious individuals.

## Figures and Tables

**Figure 1 foods-13-00198-f001:**
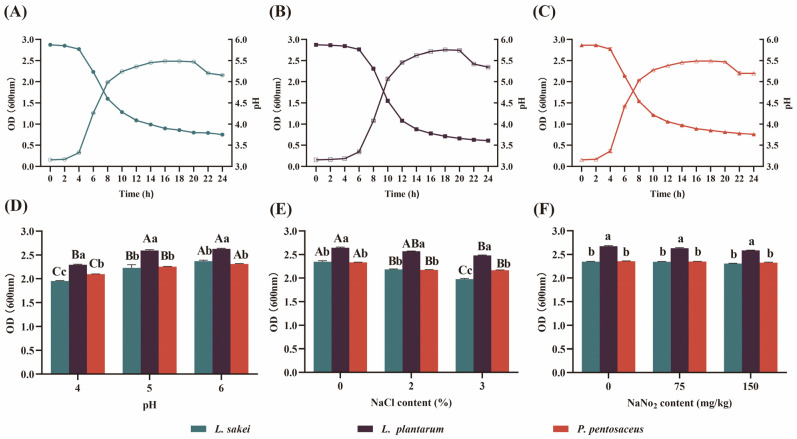
Growth potential of three LAB strains in vitro. Growth activities and acid production capacities of *L. sakei* (**A**), *L. plantarum* (**B**), and *P. pentosaceus* (**C**). Tolerance of LAB at different pH (**D**), NaCl content (**E**), and NaNO_2_ content (**F**). Different uppercase letters (A–C) mean significant differences among the three different LAB within the same condition (*p* < 0.05). Different lowercase letters (a–c) mean significant differences among the different conditions of the same LAB (*p* < 0.05).

**Figure 2 foods-13-00198-f002:**
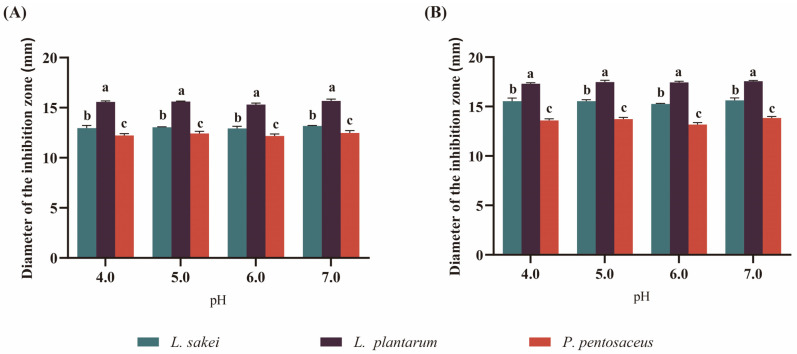
Antibacterial activity of three LAB on *E. coli* (**A**) and *S. aureus* (**B**) at different pH values. Different lowercase letters (a–c) indicate significant differences among the three different LAB within the same pH condition (*p* < 0.05).

**Figure 3 foods-13-00198-f003:**
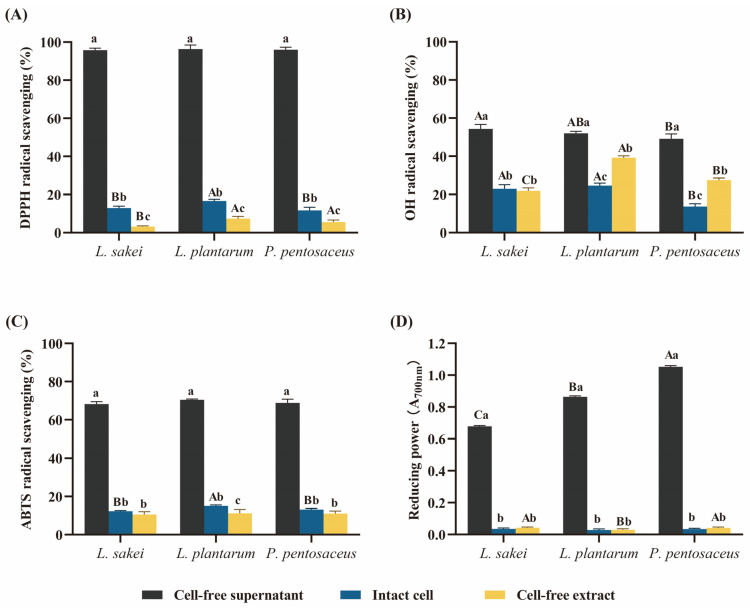
Antioxidant potential of three LAB. (**A**) 2,2-Diphenyl-1-picrylhydrazyl (DPPH), (**B**) hydroxyl (OH) radical scavenging activity, (**C**) 2,2′-azino-bis (3-ethylbenzothiazoline-6-sulfonic acid) (ABTS), and (**D**) reducing power. Different uppercase letters (A–C) mean significant differences among the three different LAB of the same type (*p* < 0.05). Different lowercase letters (a–c) mean significant differences among the three different types of the same LAB (*p* < 0.05).

**Figure 4 foods-13-00198-f004:**
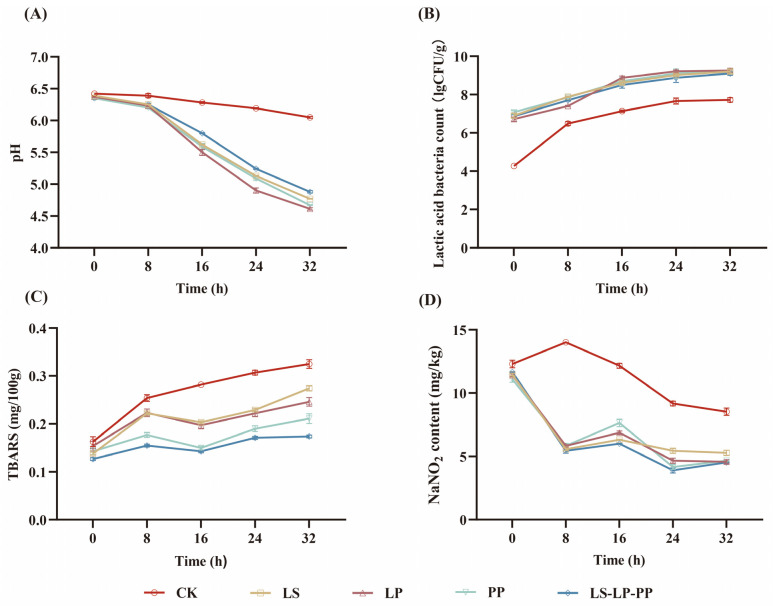
The pH (**A**), LAB counts (**B**), TBARS (**C**), and NaNO_2_ content (**D**) during fermentation.

**Table 1 foods-13-00198-t001:** Various seasonings of fermented sausage.

Seasonings	Content (*w*/*w*)
Sodium nitrite	0.01%
Sodium tripolyphosphate	0.2%
Salt	2%
Glucose	1%
Sucrose	1%
White pepper powder	0.2%

**Table 2 foods-13-00198-t002:** Scoring criteria for sensory evaluation of fermented sausages.

Evaluation Criteria (Weight)	Standard	Score
Color (20%)	The surface and cut surface of the meat filling are glossy, with a rosy color	15–20
	The surface and cut surface of the meat filling are glossy, with a light pink color	10–15
	The surface and cut surface of the meat filling is brown	5–10
	The surface and cut surface of the meat filling is dull and lacks gloss	<5
Aroma (20%)	Distinctive aroma of fermented sausages, rich and pure	15–20
	Slightly strong fermented aroma	10–15
	No noticeable fermented aroma	5–10
	Weak aroma with an unpleasant odor	<5
Texture (30%)	The cut surface of the meat filling is dense, lean and fat are tightly combined, and the boundaries are clear	25–30
	The cut surface of the meat filling is slightly loose	20–25
	The cut surface of the meat filling is loose, and the combination of lean and fat is not tight	10–20
	The cut surface is completely loose, with softening in the center	<10
Taste (30%)	Pure sour taste, full aftertaste, strong aroma, nonirritating flavor	25–30
	Mild sour taste with a slight aroma after eating	20–25
	Mild or too strong sour taste, impure fermented taste, and no lingering aroma after eating	10–20
	Excessive sour taste and improper flavor	<10

**Table 3 foods-13-00198-t003:** Antioxidant enzyme activities of three LAB.

Subjects		*L. sakei*	*L. plantarum*	*P. pentosaceus*
SOD activity (U/mL)	Cell-free supernatant	82.11 ± 2.74 ^Aa^	77.07 ± 1.36 ^Ba^	77.86 ± 1.89 ^Ba^
	Intact cell	nd	nd	nd
	Cell-free extract	13.48 ± 0.19 ^Bb^	10.88 ± 0.17 ^Cb^	14.54 ± 0.74 ^Ab^
GSH-PX activity (U/mL)	Cell-free supernatant	nd	nd	nd
	Intact cell	nd	nd	nd
	Cell-free extract	27.54 ± 1.97 ^A^	13.12 ± 1.14 ^C^	22.30 ± 1.14 ^B^
CAT activity (U/mL)	Cell-free supernatant	1.01 ± 0.05 ^Ba^	1.08 ± 0.05 ^Bb^	1.25 ± 0.07 ^Ab^
	Intact cell	0.99 ± 0.09 ^Ba^	1.45 ± 0.16 ^Aa^	1.51 ± 0.11 ^Aa^
	Cell-free extract	0.15 ± 0.03 ^Bb^	0.29 ± 0.03 ^Ac^	0.27 ± 0.05 ^Ac^

Results are expressed as means value ± standard deviation (n = 3). ^A–C^ means significant differences among the three different LAB of the same type (*p* < 0.05). ^a–c^ means significant differences among the three different types of the same LAB (*p* < 0.05). nd means not detected.

**Table 4 foods-13-00198-t004:** Color test and texture profile analysis of fermented sausages.

Group	CK	LS	LP	PP	LS-LP-PP
L*	56.22 ± 0.25 ^b^	61.02 ± 0.35 ^a^	61.61 ± 0.21 ^a^	60.65 ± 0.23 ^a^	61.32 ± 0.45 ^a^
a*	18.47 ± 0.29 ^e^	18.98 ± 0.02 ^d^	21.45 ± 0.54 ^a^	19.88 ± 0.08 ^b^	19.27 ± 0.16 ^c^
b*	12.50 ± 0.14 ^a^	11.57 ± 0.10 ^b^	9.83 ± 0.65 ^c^	10.03 ± 0.60 ^c^	10.50 ± 0.14 ^c^
Hardness (g)	93.20 ± 2.54 ^e^	250.32 ± 8.73 ^d^	911.93 ± 15.48 ^a^	501.40 ± 9.86 ^c^	604.51 ± 12.93 ^b^
Springiness	0.79 ± 0.05 ^b^	0.92 ± 0.03 ^a^	0.91 ± 0.01 ^a^	0.90 ± 0.03 ^a^	0.91 ± 0.05 ^a^
Cohesiveness	0.40 ± 0.04 ^d^	0.57 ± 0.03 ^c^	0.77 ± 0.01 ^a^	0.71 ± 0.04 ^b^	0.66 ± 0.02 ^b^
Chewiness (g)	35.70 ± 3.88 ^d^	119.89 ± 4.64 ^c^	636.61 ± 12.17 ^a^	327.14 ± 13.15 ^b^	331.01 ± 1.55 ^b^

Means ± standard deviation. ^a–e^ the different lowercase letters within a row indicate significant differences, *p* < 0.05.

**Table 5 foods-13-00198-t005:** Sensory scores of fermented sausages.

Group	CK	LS	LP	PP	LS-LP-PP
Color	14.60 ± 1.14	15.80 ± 0.84	16.40 ± 1.53	15.60 ± 1.52	15.80 ± 1.10
Odor	14.40 ± 1.11	14.20 ± 1.92	15.00 ± 1.58	14.20 ± 1.64	13.80 ± 1.79
Texture	19.80 ± 1.92 ^b^	20.40 ± 2.07 ^ab^	22.60 ± 1.14 ^a^	20.00 ± 1.58 ^ab^	20.20 ± 1.30 ^ab^
Taste	21.00 ± 2.24	21.60 ± 1.52	22.20 ± 2.59	21.00 ± 1.87	20.40 ± 1.67
Total	70.20 ± 2.59 ^b^	72.00 ± 2.92 ^ab^	76.20 ± 5.76 ^a^	70.80 ± 1.30 ^ab^	69.80 ± 3.56 ^b^

Means values ± standard deviation; ^a,b^ the different lowercase letters within a row indicate significant differences, *p* < 0.05.

## Data Availability

The original contributions presented in the study are included in the article, further inquiries can be directed to the corresponding author.

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
