# Peer review of "Characteristics of Lactic Acid Bacteria as Potential Probiotic Starters and Their Effects on the Quality of Fermented Sausages"

_foods, 2024, doi:10.3390/foods13020198_

Round 1
Reviewer 1 Report
Comments and Suggestions for Authors
The article is relevant and interesting because it examines the method of introducing probiotic microorganisms into the composition of sausages and, thus, the possibility of a positive impact on human health through innovative food products.
The effectiveness of sausage fermentation was assessed by the dynamics of the number of lactic acid microorganisms, the ability to accumulate lactic acid, as well as such important parameters as antimicrobial activity and the ability to trap free radicals. And in all respects, positive results were obtained from the use of probiotics in fermented sausages.
There are several points that I would like to draw the authors' attention to:
1. Table 3 presents data on the antioxidant activity of enzymes of lactic acid bacteria. It is indicated that SOD activity is very high for the cell-free supernatant, manifested in the cell-free extract, but not detected in intact cells. How is this possible? Similarly for the activity of GSH-PX - only the cell-free extract exhibits activity, while intact cells and the cell-free extract do not have such activity. Please make educated guesses to explain your results.
2. In Figure 4 C and D, an interesting behavior of the graphs is observed for a fermentation duration of 16 hours: the TBARS indicator increases up to 8 hours, then decreases until 16 hours and then increases again. At the same time, the sodium nitrite content decreases up to 8 hours of fermentation, then increases until 16 hours, and then decreases again. How can this be explained?
3. The results obtained in the study indicate the advisability of using probiotic microorganisms in fermented sausages. But probiotics must withstand the conditions of the gastrointestinal tract - in the stomach the acidity of the environment is very high (about 3 pH units), and in the small intestine it is slightly alkaline (about 7.5 pH units). Has the survival of probiotics been assessed in model media under such changing conditions, simulating the presence of the product in the gastrointestinal tract? Reasonably predict the influence of the human digestive system on the safety of the studied probiotics.
Overall, the article is written in clear language and well illustrated.
Reviewer 2 Report
Comments and Suggestions for Authors
General/major comments
In this paper, the authors present a study that investigate the characteristics of lactic acid bacteria as potential probiotic starters into fermented sausages. The study is interesting although it remains quite descriptive.
The materials and methods section is complete and describes well all the techniques used. The presentation of the results is not always very clear, in particular the histograms in Figure 1 are difficult to interpret, the legend (A-C) and (a-c) is not clear at all. Same thing for figures 2 and 3. The authors should modify these points to make their paper clearer.
Specific comments
Please use the current name of Lactobacillus plantarum i.e., Lactiplantibacillus plantarum.
L92, 97, 98, 103: In the materials and methods section please specify (v/v) : 1 % (v/v), etc.
Table 1: specify (w/v)
L191: change "was" to "were"
L253: Change "faster" to "fast"
Reviewer 3 Report
Comments and Suggestions for Authors
Manuscript foods-2793493
The work concerns important issue which is application of probiotic starter culture in fermented meat products.
It should be clarified whether Lactobacillus sakei, Lactobacillus plantarum, and Pediococcus pentosaceus used are probiotics. If so, the letter-number designation of the strain should be specified in the text.
Comments:
p.2.7.
What was the concentration of bacteria [ml kg -1] in starter culture added to the meat matrix? Whether the procedure used allows for introduction of the starter culture bacteria into the meat in food matrix (without introducing microbiological matrix?).
Please explain whether the procedure used allows for the introduction of starter culture bacteria into the meat in the form of a nutritional matrix. If so please specify what kind of nutritional matrix was used.
Please provide information whether the sausages were produced in laboratory or under industrial conditions.
line: 177. What kind of pork muscle was used to produce sausages? It should be outlined in the manuscript.
Line 217: Add definitions of hardness, springiness, cohesiveness, and chewiness to the text.
